# Correlates of Pancreatic Enzyme Replacement Therapy Intake in Adults with Cystic Fibrosis: Results of a Cross-Sectional Study

**DOI:** 10.3390/nu14071330

**Published:** 2022-03-22

**Authors:** Mette F. Olsen, Maria S. Kjøller-Svarre, Grith Møller, Terese L. Katzenstein, Bibi U. Nielsen, Tacjana Pressler, Jack I. Lewis, Inger H. Mathiesen, Christian Mølgaard, Daniel Faurholt-Jepsen

**Affiliations:** 1Cystic Fibrosis Centre, Department of Infectious Diseases, Rigshospitalet, 2100 Copenhagen, Denmark; terese.katzenstein@regionh.dk (T.L.K.); bibi.uhre.nielsen.01@regionh.dk (B.U.N.); tacjana.pressler@regionh.dk (T.P.); inger.hee.mabuza.mathiesen@regionh.dk (I.H.M.); daniel.faurholt-jepsen@regionh.dk (D.F.-J.); 2Department of Nutrition, Exercise and Sports, University of Copenhagen, 1958 Frederiksberg, Denmark; diaetistmariaks@gmail.com (M.S.K.-S.); gmp@nexs.ku.dk (G.M.); jack.lewis@nexs.ku.dk (J.I.L.); cm@nexs.ku.dk (C.M.); 3Paediatric Nutrition Unit, Department of Paediatric and Adolescent Medicine, Rigshospitalet, 2100 Copenhagen, Denmark

**Keywords:** cystic fibrosis, gastrointestinal symptoms, pancreatic enzyme replacement therapy, pancreatic insufficiency

## Abstract

Most people with cystic fibrosis (pwCF) develop pancreatic insufficiency and are treated with pancreatic enzyme replacement therapy (PERT). We aimed to describe the use of PERT and assess the correlates of PERT dose in adult pwCF. In a cross-sectional study at the Copenhagen CF Centre, the participants reported PERT intake, gastrointestinal (GI) symptoms and the use of concomitant treatments. Demographic and clinical characteristics were extracted from the Danish CF Registry. We used linear regression to assess the correlates of PERT dose per kg bodyweight (U-lipase/kg). We included 120 pwCF with a median age of 32.9 years, 46% women and 72% F508delta homozygote. The PERT dose ranged from 0 to 6160 U-lipase/kg per main meal (mean 1828; SD 1115). The PERT dose was associated with participants’ sex (men vs. women: 661; 95% CI: 302; 1020 U-lipase/kg), age (−16; 95% CI: −31; −1 U-lipase/kg per year) and weight (−45; 95% CI: −58; −31 U-lipase/kg per kg). Having less frequent constipation and being lung transplanted were also associated with a higher PERT dose. A third of participants did not take PERT for snacks, and this was associated with the frequency of diarrhoea. These findings indicate that PERT intake may be improved to reduce GI symptoms.

## 1. Introduction

Malnutrition is common in people with cystic fibrosis (pwCF), and nutritional status is strongly associated with pulmonary function and survival [1]. A defect in the cystic fibrosis transmembrane conductance regulator (CFTR) causes the reduced transport of chloride across cell membranes, which results in thickened mucus that obstructs the pancreatic ducts responsible for the secretion of pancreatic enzymes to the intestinal lumen [2]. The damage to the pancreas starts in utero and progresses through the first years of life [3], resulting in pancreatic exocrine insufficiency in more than 80% of adult pwCF [4].

The impaired digestion in combination with increased metabolic demands may lead to malnutrition and vitamin deficiencies in pwCF [2]. Pancreatic enzyme replacement therapy (PERT) is standard nutritional care for pwCF with pancreatic insufficiency [5]. The diagnosis of pancreatic insufficiency is defined as the absence of pancreatic enzymes in two stool samples or elevated levels of fat in stools, but the initiation of PERT is often based on the clinical judgement of stool patterns and gastrointestinal (GI) symptoms [4,6].

Abdominal pain and other GI symptoms are common complaints among both children and adults with CF [7,8]. The GI manifestations of CF have been less studied than the pulmonary consequences of the disease [9] and abdominal symptoms are often undertreated [10]. Although the safety and effectiveness of PERT are well established [11,12], there is a lack of knowledge about the use of PERT, including the optimal dose and timing of PERT intake in pwCF [11,13,14].

At the Copenhagen CF Centre in Denmark, pwCF are administered PERT as enteric-coated formulation (mainly Creon^®^ 10.000 and Creon^®^ 25.000) based on their report of GI symptoms or a faecal elastase-1 < 200 mcg/g. Adult pwCF are counselled to take 1.000–3.000 U lipase/kg bodyweight for their main meals. Less PERT may be needed for smaller meals and snacks. PwCF with a higher bodyweight are counselled to use less PERT per kg, since they are likely to have relatively more metabolic inactive fat mass and since the relative fat content of meals is expected to be lower in larger individuals. Furthermore, patients are advised to adjust their dose according to the size and fat content of each meal, as well as stool frequency and GI symptoms [15,16]. The adjustment of the PERT dosage is done in consultation with dietitians at the CF Centre, but in contrast to antibiotics and other CF treatments, the intake of PERT is mainly managed by the patients themselves outside the care of the CF team. If the PERT dose is inadequate, it may not be detected and corrected, as it may be difficult for patients to identify whether GI symptoms require adjustments of their PERT intake.

We recently studied the PERT intake in our paediatric CF population and found that GI symptoms were common and did not improve by changing the timing of the PERT intake from before to after meals [17]. In the present study, we aimed to describe use of PERT and GI symptoms among adults with CF and to assess the associations between PERT dose and the demographic and clinical characteristics of pwCF.

## 2. Materials and Methods

We conducted a cross-sectional study among pwCF at the Copenhagen CF Centre at Rigshospitalet, Denmark. We invited a convenience sample of all patients presenting for outpatient consultations in May and June 2020. The inclusion criteria were (1) ≥18 years of age, (2) confirmed CF diagnosis, either by two CF-causing mutations in the CFTR gene and/or elevated sweat chloride, and (3) consenting to participate.

A questionnaire was developed for the study to obtain information about the dose of PERT for main meals and snacks, the timing of PERT intake (before, during or after meals) and reasons for changing the PERT dose. Participants reported GI symptoms including abdominal pain, constipation and diarrhoea, stool frequency and stool consistency using the Bristol stool scale [18]. Further questions included the use of laxatives, acid suppressants, vitamin supplements, and ursodeoxycholic acid. The questionnaire was administered as an interview during outpatient consultations and the data were recorded using REDCap (Research Electronic Data Capture).

In addition, data were extracted from the Danish Cystic Fibrosis Registry to obtain information about the demographic and clinical characteristics of all adult pwCF followed at the Copenhagen CF Centre to enable a comparison between the study participants and non-participants. The extracted data included age, sex, weight, BMI, CFTR mutation, CFTR modulator therapy, faecal elastase-1, lung function (assessed by spirometry and expressed as the percent of predicted forced expired volume in 1 s, ppFEV1 [19]), lung transplantation status, chronic lung infection status and comorbidities, including diabetes and liver disease (assessed by ultrasound and defined as cirrhosis, fibrosis or steatosis) and vitamin D status (serum 25-hydroxyvitamin D). Pancreatic insufficiency was defined as faecal elastase-1 < 200 mcg/g [4].

The data analyses were carried out using Stata/SE version 17.0 (StataCorp LP, College Station, TX, USA). The data are presented as mean (SD), median (IQR) or *n* (%) for normal, non-normal and categorical variables, respectively. We compared the characteristics of the participants and non-participants using a Chi-square test, Student’s *t*-test and Kruskal–Wallis test, as appropriate. The PERT dose was calculated as lipase units per kg bodyweight (U lipase/kg). The correlates of the PERT dose per main meal were assessed using linear regression, and the estimates are presented unadjusted as well as adjusted for sex, age and weight. Linear regression was also used to assess any potential associations between the timing of PERT intake and GI symptoms. Lastly, we used logistic regression to assess whether use of PERT for snacks was associated with patient characteristics or GI symptoms. Associations with *p*-values < 0.05 were considered significant.

## 3. Results

We invited 120 pwCF who attended outpatient consultations in May and June 2020 to participate in the study, and all consented. The demographic and clinical data were extracted for a total of 211 adult pwCF at the Copenhagen CF Centre. The characteristics of the study participants and non-participants are presented in Table 1. The participants had a median age of 32.9 years (range: 18–60), 46% were women and the majority (72%) were homozygous for F508 delta. Their mean (SD) BMI was 22.7 (4.0) kg/m^2^, 8% were underweight, while 17% were overweight. Among those overweight, six participants were obese (BMI > 30) while none were severely obese (BMI > 40). None of the participants required gastronomy tube feeding.

Ninety-one percent of the participants and 80% of non-participants had pancreatic insufficiency (*p* = 0.052). The patients’ CF genotype was associated with pancreatic insufficiency, and those homozygous for F508 delta had faecal elastase-1 < 200. The study participants were representative of the adult patient population at the CF Centre in terms of sex, age, nutritional status, chronic lung infections and comorbidities. Compared to the non-participants, more study participants were homozygous for F508 delta (81% vs. 53%, *p* = 0.003) and had poorer lung function (−9 ppFEV1, *p* = 0.01), fewer were lung transplanted (7% vs. 14%, *p* = 0.02) and they had a higher mean vitamin D level (+12 nmol/L, *p* = 0.01) compared to the pwCF not included in the survey.

The study participants reported PERT doses for main meals ranging from 0 to 6160 U lipase/kg with a mean intake of 1828 (1115) U lipase/kg (Table 2, Figure 1). Eight participants did not use PERT. Of these, seven were pancreatic sufficient. One participant did have pancreatic insufficiency (faecal elastase-1 < 15 mcg/g), but reported no current use of PERT. Among those taking PERT, three participants were found not to have pancreatic insufficiency (faecal elastase-1: 289, 487 and 489 mcg/g, respectively).

A third of the participants did not take any PERT for snack meals, while the others reported up to 6160 U lipase/kg per snack (median 702; IQR 0–1517). Most participants took PERT before (73%) or during (20%) meals. The majority reported that they adjusted their PERT dose according to the fat content and size of their meals, while fewer adjusted their dose if having gastrointestinal symptoms and a quarter of participants did not adjust their dose.

Weekly abdominal pain was reported by 53 participants (44%), of whom 10 experienced symptoms every day (Table 3). Constipation and diarrhoea were also common GI symptoms, with 26% and 45%, respectively, reporting to have symptoms one or more days per month. At the time of the interview, 11% had some degree of constipation, while 23% reported some degree of diarrhoea. A fifth of the participants used laxatives. Almost half used proton pump inhibitors or other acid suppressants. The majority of the participants took fat-soluble vitamins (A–D–E–K) combinations and/or other micronutrient supplements, and a third used ursodeoxycholic acid.

The PERT dose taken for main meals was associated with participants’ sex, age and weight. Men took more PERT for main meals than women: 661 (95% CI: 302; 1020) U lipase/kg (Table 4). Age and weight were inversely associated with PERT intake: −16 (95% CI: −31; −1) U lipase/kg per year of age and −45 (95% CI: −58; −31) U lipase/kg per kg bodyweight. The association with body size was also seen from the BMI: the PERT dose among underweight participants was 657 (95% CI: −58; 1373) U lipase/kg higher, while the dose among those overweight was 983 (95% CI: 484; 1481) U lipase/kg lower, compared to participants with normal weight (BMI 18.5–25).

In addition, the PERT dose was associated with GI symptoms. Participants with constipation for one or more days/month used 615 (95% CI: 228; 1002) U lipase/kg less than participants who never experienced constipation. Mild/severe constipation and diarrhoea were both associated with marginally lower PERT doses than those reporting normal stool consistency: −541 (95% CI: −1088; 6) U lipase/kg and −407 (95% CI: −816; 2) U lipase/kg, respectively. Abdominal pain was not associated with PERT dose.

Seven participants were lung transplanted. Their PERT dose was 1206 (95% CI: 490; 1921) U lipase/kg higher than that of other participants. Furthermore, CF-related diabetes was associated with a marginally higher PERT dose: 370 (95% CI: −8; 748) U lipase/kg. We did not find that PERT use for main meals was associated with CF-related liver disease, vitamin D status, use of concomitant treatment or severity of CF disease (CF mutation, lung function or chronic lung infection). Lastly, we did not find any associations between treatment with CFTR modulators and PERT intake or between the timing of PERT intake and GI symptoms (data not shown).

The participants who did not use PERT for snacks also used less PERT for their main meals: −679 (95% CI: −1041; −317) U lipase/kg. Men were more likely than women to use PERT for snack meals: odds ratio 3.6 (95% CI: 1.3; 9.5) and lack of PERT intake for snacks was associated with more frequent diarrhoea: 2.3 (95% CI: 0.2; 4.5) days/month after adjusting for sex, age and weight.

## 4. Discussion

In this study, we found a high burden of GI symptoms and a large variation in the PERT dose used by adult pwCF. Male sex, younger age, lower bodyweight, less frequent constipation and being lung transplanted were the factors associated with using a higher dose of PERT/kg. PERT intake was not associated with abdominal pain, stool patterns or the severity of CF disease. A large proportion of pwCF, especially women, did not take any PERT for snacks and we found that this practice was associated with more frequent diarrhoea. In addition, we did not find that the timing of PERT intake was associated with GI symptoms.

Our findings are in line with previous studies among paediatric CF populations, which also found large variations in PERT doses [14]. The high variability points to the lack of a general criterion to adjust PERT and a lack of direct methods to assess individual PERT requirements. The adequacy of the PERT dosage is mainly evaluated by the patients themselves based on stool patterns and GI symptoms. The high frequency of GI symptoms in our study may indicate a potential for improving the guidance of patients to optimise PERT intake. In paediatric CF populations, the introduction of a self-management mobile app to support nutritional intake and the individual adjustment of PERT has shown promising effects on PERT adherence and improved GI functioning [20].

In addition, we saw a mismatch between pancreatic insufficiency and PERT use: one participant was not using PERT despite being pancreatic insufficient, and three were taking PERT despite being pancreatic sufficient based on the analyses of faecal elastase-1. In the latter cases, PERT had possibly been administered based on GI symptoms before faecal elastase-1 was measured. This highlights the risk of excessive use when PERT is administered based on GI symptoms. A study in the US found that more than half of pwCF with faecal elastase-1 > 200 mcg/g had been incorrectly classified as pancreatic insufficient based on their GI symptoms, and were thus inappropriately treated with PERT [6]. The measurement of faecal elastase-1 for the screening of pancreatic status is thus important to avoid adding unnecessary therapies to the already extensive treatment burden in the CF population.

We also found a high burden of GI symptoms, in line with observations from previous studies in both adult and paediatric CF populations [21,22]. We saw that lower PERT doses were associated with more frequent constipation. Although we cannot conclude on the causality of this association, it is likely that low doses of PERT may have contributed to constipation. Similar to a previous study in both paediatric and adult pwCF, we did not find the PERT dose to be associated with abdominal pain or stool patterns [23]. This suggests that these variables may not be good clinical indicators for adjustments of PERT doses. The underlying causes of GI symptoms in pwCF are complex, and the need for a better understanding of the optimal PERT doses and more sensitive indicators for dose adjustments merit further research [9,11].

Our findings show that those more inclined to use a higher PERT dose in general are men, younger and have lower weight. This is useful information for clinicians and dieticians counselling pwCF on PERT use, which has not been documented previously. Men took more PERT for main meals and they were also more likely to take PERT for snacks than women. This observed sex difference was not explained by variation in age or weight and merits further investigation. The fact that older pwCF used a lower PERT dose than younger individuals may be explained by habit due to changes in the effectiveness of PERT products and in the guidance given by CF dieticians over time [5]. Lastly, the inverse association seen between the PERT dose per kg bodyweight and participants’ weight is in agreement with nutritional guidelines [15,16]. The decrease in PERT intake with increasing weight is likely to be a result of both dietician counselling and pwCF’s own adjustments.

We also found an unexpected association between PERT dose and lung transplantation, showing that pwCF with lung transplants used a higher PERT dose. However, this finding should be interpreted with caution, as it is based on few participants (*n* = 7), who are not likely to represent the full sample of CF patients with lung transplants.

A surprisingly high proportion of pwCF, especially women, did not take PERT for snack meals. The need for enzyme replacement for a snack depends on its fat content, but may be underestimated for snacks such as nuts and energy bars. We found that pwCF who did not use PERT for snacks also used a lower dose for main meals, and found that they had a higher frequency of diarrhoea.

Lastly, we did not find that the timing of PERT intake was associated with GI symptoms and thus cannot further elucidate the question of optimal timing. We recently tested changes in PERT timing in our paediatric CF population, but found no clear effects on GI symptoms. In line with this, a recent systematic review concluded that there is currently no evidence to support the best time to take PERT in relation to a meal [13].

About half of the participants in this study were treated with CFTR modulators, including nine participants who had recently initiated treatment with triple CFTR modulators. We did not find that CFTR modulators was associated with PERT use, and although it may be premature to assess the potential effects in our study population, pancreatic insufficiency in adult pwCF has generally been considered irreversible. However, emerging data in young children treated with CFTR modulators suggest some degree of recovery of the exocrine pancreatic function [24,25,26]. Modulators have also shown beneficial effects on intestinal microbiome and epithelia function [27,28]. Thus, it is likely that the future needs of PERT will change in the era of triple CFTR modulator therapy. However, enzyme replacement will continue to be an integral part of nutritional care for pwCF and there will be a continued need for patient support during future adjustments of PERT intake with changing requirements.

The strengths of this study include a relatively large sample of 120 participants and access to detailed demographic and clinical information about the full adult patient population at the Copenhagen CF Centre with very high data completeness, which showed that the study participants were representative in terms of sex, age and weight, which were the main PERT correlates identified. Several study limitations should also be noted. As the study was based on a convenience sample invited at outpatient consultations, pwCF were more likely to be included if they had more advanced CF disease and/or were more compliant with CF care. Consequently, the study participants had more severe CF mutations, poorer lung function and higher vitamin D levels (indicating better adherence to micronutrient supplementation recommendations), than those not included in the study. The participants also included fewer pwCF with lung transplantation, as these are mainly followed by transplantation specialists and seen less often at the CF Centre. These limitations of the representativeness of our sample means that we may have overestimated the PERT intake compared to the general patient population. However, we do not have reason to expect that this caused bias in our estimation of the correlates of PERT intake. In addition, our study is limited by the potential recall bias in the retrospective assessment of GI symptoms and, due to the cross-sectional nature of this study, we are not able to determine causality between PERT use and GI symptoms.

## 5. Conclusions

In conclusion, this study found a large variation in the use of PERT among adult pwCF and a high burden of GI symptoms. PERT dose was associated with patient sex, age, bodyweight, frequency of constipation and lung transplantation. A third of pwCF did not use PERT for snack meals and we found that this was associated with a higher frequency of diarrhoea. Based on these findings, it seems that an improved understanding of patients’ dosage and adjustment of PERT intake, as well as more knowledge on the optimal dose and timing of PERT, is needed to optimise counselling and improve GI function in pwCF.

## Figures and Tables

**Figure 1 nutrients-14-01330-f001:**
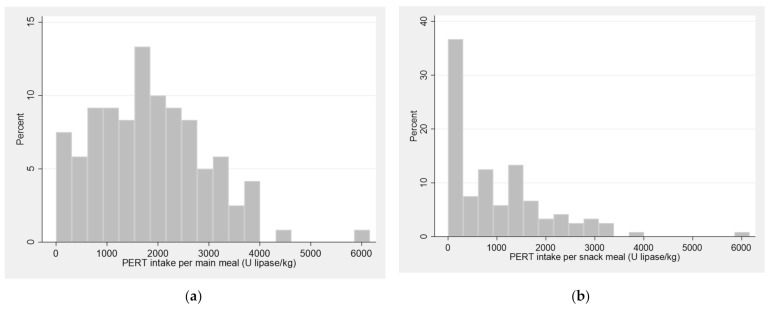
Intake of pancreatic enzyme replacement therapy for (**a**) main meals and (**b**) snack meals.

**Table 1 nutrients-14-01330-t001:** Characteristics of participants and non-participants among adults with cystic fibrosis.

	Participants (*n* = 120)	Non-Participants (*n* = 91)	*p* ^a^
Sex (female)	55 (45.8)	46 (50.6)	0.50
Age (year)	32.9 (26.1; 45.1)	31.2 (24.4; 38.7)	0.054
18–29 year	47 (39.2)	41 (45.1)	0.10
30–39 year	28 (23.3)	30 (33.0)	
40–49 year	27 (22.5)	13 (14.3)	
50+ year	18 (15.0)	7 (7.7)	
Weight (kg)	66.4 ± 13.3	66.8 ± 14.5	0.85
BMI (kg/m^2^)	22.7 ± 4.0	22.7 ± 4.0	0.98
Underweight (<18.5)	9 (7.5)	9 (9.9)	0.66
Normal weight (≥18.5 to 25)	91 (75.8)	64 (70.3)	
Overweight (≥25)	20 (16.7)	18 (19.8)	
Faecal elastase-1 ^b^			0.052
<100 mcg/g	99 (90.8)	52 (80.0)	
100–200 mcg/g	0 (0.0)	2 (3.1)	
>200 mcg/g	10 (9.2)	11 (16.9)	
CFTR mutation			0.003
F508 delta homozygote	81 (72.3)	53 (53.5)	
F508 delta heterozygote	30 (26.8)	38 (38.4)	
Other	1 (0.9)	8 (8.1)	
CFTR mutation class ^b^			0.02
I	0 (0.0)	3 (3.3)	
II	111 (96.4)	73 (81.1)	
III	1 (0.9)	1 (1.1)	
IV	4 (1.8)	9 (10.0)	
V	1 (0.9)	4 (4.4)	
CFTR modulator therapy			0.06
None	54 (45.0)	55 (60.4)	
Mono/dual CFTR modulator	57 (47.5)	33 (36.3)	
Triple CFTR modulator	9 (7.5)	3 (3.3)	
Lung function (ppFEV1) ^b^	68.1 ± 26.4	77.2 ± 26.4	0.01
Chronic lung infection	90 (75.0)	62 (68.1)	0.27
Lung transplanted	7 (5.8)	14 (15.4)	0.02
Comorbidities			
Diabetes	43 (35.8)	24 (26.4)	0.14
Liver disease ^c^	20 (16.7)	10 (11.0)	0.24
Vitamin D status (nmol/L) ^b^	82.7 (±31.8)	70.7 (±33.3)	0.01

Data are *n* (%), mean ± SD or median (IQR). BMI: body mass index; CFTR: cystic fibrosis transmembrane conductance regulator; ppFEV1: percent of predicted forced expiratory volume in 1 s. ^a^ Difference based on Chi-squared, Student’s *t*-test or Kruskal–Wallis test, respectively. ^b^ Data available for participants/non-participants: faecal elastase-1: 109/65; CFTR mutation class: 117/90; vitamin D status: 117/84. ^c^ Liver disease defined as cirrhosis, fibrosis or steatosis assessed by ultrasound.

**Table 2 nutrients-14-01330-t002:** Use of pancreatic enzyme replacement therapy in 120 adults with cystic fibrosis.

	*n* (%)
PERT dose per main meal (U lipase/kg)	
Mean ± SD	1828 ± 1115
Range	0–6160
=0	8 (6.7)
>0 to <1000	22 (18.3)
≥1000 to <2000	37 (30.8)
≥2000 to <3000	36 (30.0)
≥3000	17 (14.2)
PERT dose per snack meal (U lipase/kg)	
Median (IQR)	702 (0–1517)
Range	0–6160
=0	40 (33.3)
>0 to <1000	31 (25.8)
≥1000 to <2000	31 (25.8)
≥2000 to <3000	12 (10.0)
≥3000	6 (5.0)
Timing of PERT intake ^a^	
Before meals	82 (73.2)
During meals	22 (19.6)
After meals	8 (7.1)
Reasons to change PERT dose ^a,b^	
Fat content of meal	23 (27.4)
Size of meal	12 (14.3)
Gastrointestinal symptoms	8 (9.5)
Combinations of the above	20 (24.4)
Does not change PERT dose	21 (25.0)

Data are mean ± SD, median (IQR) or *n* (%). PERT: pancreatic enzyme replacement therapy, lipase units per kg bodyweight (U lipase/kg). ^a^ Among those taking PERT. ^b^ Data available for 84 participants.

**Table 3 nutrients-14-01330-t003:** Gastrointestinal symptoms and use of laxatives, acid suppressants, vitamin supplements and ursodeoxycholic acid in 120 adults with cystic fibrosis.

	*n* (%)
Abdominal pain	
Never	67 (55.8)
Once a week	30 (25.0)
Several days a week	13 (10.8)
Every day	10 (8.3)
Constipation (days/month)	
0	89 (74.2)
1–2	20 (16.7)
3–14	8 (6.7)
15–30	3 (2.5)
Diarrhoea (days/month)	
0	66 (55.0)
1–2	28 (23.3)
3–14	21 (17.5)
15–30	5 (4.2)
Stool frequency/day	
≤1	34 (28.3)
>1–2	53 (44.2)
>2–3	27 (22.5)
>3	6 (5.0)
Bristol stool chart	
Severe constipation	3 (2.5)
Mild constipation	10 (8.3)
Normal	79 (65.8)
Mild diarrhoea	19 (15.8)
Severe diarrhoea	9 (7.5)
Laxative use (yes)	23 (19.2)
Antacid use	
Proton pump inhibitors	45 (37.5)
Other acid suppressants	7 (5.8)
No acid suppressants used	68 (56.7)
Vitamin supplement use	
A–D–E–K vitamin combination	95 (79.2)
Vitamin D and calcium	50 (41.7)
Multivitamin and/or other supplements	47 (39.2)
No vitamin supplements used	9 (7.5)
Ursodeoxycholic acid use (yes)	41 (34.2)

Data are *n* (%).

**Table 4 nutrients-14-01330-t004:** Correlates of PERT intake per main meal in 120 adults with cystic fibrosis.

	U lipase/kg, Mean (95% CI) Unadjusted	*p* ^a^	U lipase/kg, Mean (95% CI) Adjusted *	*p* ^a^
Sex				
Female	Ref		Ref	
Male	304 (−99; 706)	0.14	661 (302; 1020)	<0.001
Age (year)	−18 (−36; −0.4)	0.045	−16 (−31; −1)	0.041
18–29 y	Ref	0.07	Ref	0.08
30–39 y	43 (−476; 561)		−23 (−468; 422)	
40–49 y	−595 (−1119; −71)		−522 (−972; −73)	
50+ y	−421 (−1023; 181)		−387 (−902; 128)	
Weight (kg)	−37 (−51; −24)	<0.001	−45 (−58; −31)	<0.001
BMI (kg/m^2^)	−123 (−169; −77)	<0.001	−118 (−164; −73)	<0.001
Underweight (<18.5)	791 (78; 1504)	0.03	657 (−58; 1373)	0.07
Normal weight (≥18.5 to 25)	Ref		Ref	
Overweight (≥25)	−989 (−1493; −486)	<0.001	−983 (−1481; −484)	<0.001
CFTR mutation				
F508 delta homozygote	Ref		Ref	
F508 delta heterozygote	−577 (−1005; −150)	0.01	−269 (−647; 109)	0.16
Other	−1000 (−3171; 1171)	0.36	−1388 (−3270; 493)	0.15
Lung function (ppFEV_1_)	−1 (−9; 7)	0.81	2 (−6; 10)	0.63
Lung transplanted	1384 (558; 2210)	0.001	1206 (490; 1921)	0.001
Chronic lung infection	16 (−452; 484)	0.95	−214 (−637; 209)	0.32
Comorbidities				
Diabetes	260 (−154; 674)	0.22	370 (−8; 748)	0.055
Liver disease ^b^	−29 (−572; 515)	0.92	−366 (−834; 101)	0.12
Vitamin D status (nmol/L) ^c^	−2 (−8; 5)	0.58	0.2 (−6; 6)	0.95
Abdominal pain				
≥1 days/week	61 (−346; 469)	0.77	145 (−217; 506)	0.43
Constipation				
≥1 days/month	−827 (−1265; −390)	<0.001	−615 (−1002; −228)	0.002
Diarrhoea				
≥1 days/month	−114 (−521; 292)	0.58	57 (−298; 411)	0.75
Bristol scale				
Mild/severe constipation	−615 (−1270; 39)	0.07	−541 (−1088; 6)	0.053
Normal	Ref		Ref	
Mild/severe diarrhoea	−333 (−814; 148)	0.17	−407 (−816; 2)	0.051
Concomitant treatment				
Antacid use	−210 (−617; 197)	0.31	−225 (−582; 131)	0.21
Vitamin suppl. use (A–D–E–K)	−110 (−619; 399)	0.67	298 (−127; 723)	0.17
Ursodeoxycholic acid use	456 (37; 875)	0.03	199 (−177; 574)	0.30

* Adjusted for age, sex and weight. PERT: pancreatic enzyme replacement therapy (lipase units per kg bodyweight); BMI: body mass index; CFTR: cystic fibrosis transmembrane conductance regulator; ppFEV1: percent of predicted forced expired volume in 1 s. ^a^ Mean difference based on linear regression. ^b^ Liver disease defined as cirrhosis, fibrosis or steatosis assessed by ultrasound. ^c^ Data on vitamin D available for 117 participants.

## Data Availability

The data presented in this study are available on request from the corresponding author. The data are not publicly available due to the conditions of Danish legislation in Article 10 of the Danish Act on Data Protection. It is stated in Article 10 of the said Act that personal research data can be transferred to scientific journals for verification of the research results. However, the Danish Act on Data Protection does not allow for personal data to be made available to others without prior individual approval from the Danish Data Protection Agency.

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
