# Peer review of "Correlates of Pancreatic Enzyme Replacement Therapy Intake in Adults with Cystic Fibrosis: Results of a Cross-Sectional Study"

_nutrients, 2022, doi:10.3390/nu14071330_

Round 1

Reviewer 1 Report

In this paper, the authors report the use of PERT in an adult population with CF and look for dose-related factors or correlation between dose and gastrointestinal symptoms. I found the paper of little interest. It is known that the dose of PERT can be related to gastrointestinal symptoms. The correlations with sex, age and weight are in my opinion expected and of little interest. The two groups are not comparable, as CFTR genotype and disease severity are different. The evaluation in the subgroup of transplant patients could have been interesting but it includes very few patients.

Author Response

We thank the reviewer for commenting on our manuscript. We acknowledge that there may be expectations of correlations between PERT dose and demographic/clinical characteristics based on clinical experience but, to the best of our knowledge, there is very little data documenting these associations in the scientific literature. In fact, the relations between PERT and gastrointestinal symptoms, mentioned as known by the reviewer, were not strong in our data. We did not find PERT dose to be associated with neither abdominal pain or stool patterns, which indicates that these symptoms may not be good clinical indicators for adjustment of PERT dose. Similarly, the identified correlations between PERT dose and sex, age and weight is useful information for clinicians and dieticians counselling pwCF on PERT use, which has not previously been reported in the literature. We have emphasized this in the revised manuscript (line 222-225).

The purpose of comparing the two groups in table 1 (participants and non-participants) is merely to describe how well the study participants represented the entire adult CF population at the Copenhagen CF Centre. We agree with the reviewer that there are differences in CFTR genotype and disease severity, as expected when sampling from an outpatient clinic. In our discussion of strengths and limitations, we mention that we don’t have reason to expect that these differences have caused bias in our estimation of PERT dose correlates (line 266-268). In the revised manuscript, we have added that the study participants are representative of the full CF population in terms of sex, age and weight, the main correlates of PERT dose identified (line 119-121 and line 262-264).

We agree with the reviewer that the finding among patients with lung transplant is potentially interesting but needs to be explored in a larger population.

Reviewer 2 Report

In this cross-sectional study, the authors nicely detailed the associated factors with PERT in CF adults. The results were clearly portrayed, and limitations/conclusions were appropriately detailed. I have the following minor suggestions. 

    1. Did any of the participants require Gastrostomy tube feeding?
    2.  Table 1- was there any obese (BMI >30) and morbidly obese (>40)?
    3. Table 1 - please include the status of other fat-soluble vitamins (vitamin A and E) between the two groups
    4. Obstipation - please define this in detail. How is obstipation different from severe constipation and distal intestinal obstruction syndrome?
    5. Table 3 - antacids - Referring to this as "acid suppressants" will be more appropriate. 

Author Response

Thank you for the positive review. Please find our responses to the specific comments below:

  • None of the participants required tube feeding. We have added this information to the revised manuscript. (line 108-109)
  • A total of 12 pwCF were obese (6 participants and 6 non-participants). None had a BMI >40. We have added this to the text referring to table 1 and deleted a few repetitions from the table (line 106-108)
  • We do not have data on other fat-soluble vitamins since only vitamin D is assessed routinely.
  • In the revised manuscript, we have replaced “obstipation” with “constipation”, as this is a more correct English translation of the Danish term used in the questionnaire.
  • We agree with the reviewer and have replaced “antacids” with “acid suppressants” in table 3 and text.

Round 2

Reviewer 1 Report

I really appreciate the efforts made by the authors in order to improve the work. The data presented is useful for clinicians. It is true that what has been demonstrated in the text can be hypothesized in clinical practice. Nevertheless, little is actually described in the literature on the subject. However, I recommend to specify well that the genotype of patients can be correlated with the dose of PERT required. For example, from the genotype it is possible to predict the risk of pancreatic insufficiency, both in pediatric age (see Terlizzi V et al. JCF 2014) and in adulthood (see Ooi et al Gastroenterology 2011). It is necessary to specify this aspect better. I have nothing else to add.

Author Response

Thanks for the additional comment on genotype. We agree with the reviewer that the genotype of CF patients is associated with their risk of pancreatitis and the risk of pancreatic insufficiency (as reported in the two mentioned references). The majority of our patient population is F508 delta homozygote (Table 1) and they have a higher prevalence of pancreatic insufficiency than those who are F508 delta heterozygote. We have added this observation to our result section (line 118-119). However, within those with pancreatic insufficiency, it is not known if the genotype can be correlated with the PERT dose required. In our data, PERT intake was not associated with CFTR mutation or other indicators of the severity of CF disease, incl. lung function or chronic lung infection (line 172-173 + line 192-193).